# Effects of Consuming Sugar-Sweetened Beverages for 2 Weeks on 24-h Circulating Leptin Profiles, Ad Libitum Food Intake and Body Weight in Young Adults

**DOI:** 10.3390/nu12123893

**Published:** 2020-12-19

**Authors:** Desiree M. Sigala, Adrianne M. Widaman, Bettina Hieronimus, Marinelle V. Nunez, Vivien Lee, Yanet Benyam, Andrew A. Bremer, Valentina Medici, Peter J. Havel, Kimber L. Stanhope, Nancy L. Keim

**Affiliations:** 1Department of Molecular Biosciences, School of Veterinary Medicine, University of California Davis, Davis, CA 95616, USA; dmsigala@ucdavis.edu (D.M.S.); bettina.hieronimus@mri.bund.de (B.H.); mvnunez@ucdavis.edu (M.V.N.); vilee@ucdavis.edu (V.L.); ybenyam@ucdavis.edu (Y.B.); pjhavel@ucdavis.edu (P.J.H.); 2Department of Nutrition, University of California Davis, Davis, CA 95616, USA; nancy.keim@ars.usda.gov; 3Department of Nutrition, Food Science, and Packaging, San Jose State University, San Jose, CA 95192, USA; adrianne.widaman@sjsu.edu; 4Institute for Physiology and Biochemistry of Nutrition, Max Rubner-Institut, 76131 Karlsruhe, Germany; 5Department of Pediatrics, School of Medicine, University of California Davis, Sacramento, CA 95817, USA; andrew.bremer@nih.gov; 6Department of Internal Medicine, Division of Gastroenterology and Hepatology, University of California Davis, Sacramento, CA 95817, USA; vmedici@ucdavis.edu; 7Department of Basic Sciences, Touro University of California, Vallejo, CA 94592, USA; 8Western Human Nutrition Research Center, United States Department of Agriculture, Davis, CA 95616, USA

**Keywords:** leptin, satiety, energy intake, energy compensation, obesity, aspartame, fructose, glucose, sucrose, high fructose corn syrup

## Abstract

Sugar-sweetened beverage (sugar-SB) consumption is associated with body weight gain. We investigated whether the changes of (Δ) circulating leptin contribute to weight gain and ad libitum food intake in young adults consuming sugar-SB for two weeks. In a parallel, double-blinded, intervention study, participants (*n* = 131; BMI 18–35 kg/m^2^; 18–40 years) consumed three beverages/day containing aspartame or 25% energy requirement as glucose, fructose, high fructose corn syrup (HFCS) or sucrose (*n* = 23–28/group). Body weight, ad libitum food intake and 24-h leptin area under the curve (AUC) were assessed at Week 0 and at the end of Week 2. The Δbody weight was not different among groups (*p* = 0.092), but the increases in subjects consuming HFCS- (*p* = 0.0008) and glucose-SB (*p* = 0.018) were significant compared with Week 0. Subjects consuming sucrose- (+14%, *p* < 0.0015), fructose- (+9%, *p* = 0.015) and HFCS-SB (+8%, *p* = 0.017) increased energy intake during the ad libitum food intake trial compared with subjects consuming aspartame-SB (−4%, *p* = 0.0037, effect of SB). Fructose-SB decreased (−14 ng/mL × 24 h, *p* = 0.0006) and sucrose-SB increased (+25 ng/mL × 24 h, *p* = 0.025 vs. Week 0; *p* = 0.0008 vs. fructose-SB) 24-h leptin AUC. The Δad libitum food intake and Δbody weight were not influenced by circulating leptin in young adults consuming sugar-SB for 2 weeks. Studies are needed to determine the mechanisms mediating increased energy intake in subjects consuming sugar-SB.

## 1. Introduction

It is anticipated that by 2030 over 86% of all Americans will be either overweight or obese [1]. While the etiology of obesity is complex, epidemiological and experimental evidence have shown that consumption of sugar-sweetened beverage (sugar-SB) increases energy intake and risk for weight gain in both youth and adults [2,3]. Studies conducted from 2011–2014 found that about one-half of US adults and about two-thirds of youth consumed at least one sugar-SB on a given day [4,5].

Given the relationship between sugar-SB consumption and weight gain, it is important to understand the mechanisms that influence positive energy intake during sustained sugar-SB consumption. Research suggests that consumption of sugar-SB may lead to incomplete energy compensation due to its failure to elicit appropriate homeostatic responses in satiety and appetite hormones [6,7]. These homeostatic hormone responses involve coordinated changes that can either suppress energy intake (glucagon-like peptide, gastric inhibitory polypeptide, peptide YY, leptin, etc.) or promote energy intake (ghrelin) by providing information to the hypothalamus and brain stem about acute nutritional status and adiposity [8]. We have hypothesized that the effect of fructose to decrease 24-h leptin concentrations may be a mechanism by which sugar-SB promotes increased energy intake and body weight gain [9]. In response to insulin-stimulated glucose uptake, leptin is secreted by adipocytes and signals vital metabolic states to the hypothalamus [10,11]. These signals help to control and regulate body weight by decreasing energy intake and increasing energy expenditure [12,13]. In support of this, an ad libitum high-carbohydrate diet decreased energy intake and body weight, and the amplitude of the leptin peak was inversely correlated to the changes of body weight and fat mass [14]. Unlike glucose, ingestion of fructose does not elicit insulin secretion and, therefore, blunts leptin response. In two crossover studies by Teff et al., it was demonstrated that acute consumption of fructose-SB with 3 isocaloric meals decreased 24-h circulating leptin response compared with the consumption of glucose-SB in normal weight females and obese males and females [15,16]. This has also been observed during sustained consumption of fructose- or glucose-SB in older overweight and obese adults (age: 54 ± 8(SD) years; body mass index (BMI) 29.1 ± 2.9(SD) kg/m^2^) [17]. However, in these subjects, the changes of body weight and ad libitum food consumption [18] were not associated with the changes of leptin levels [17]. It is possible though, that younger, leaner individuals may be more responsive to changes in circulating leptin.

To the best of our knowledge, there are no clinical trials that have investigated whether the fructose-induced reduction in day-long circulating leptin is linked to increased energy intake and body weight gain in young adults. Therefore, the objective of this study was to investigate the effects of consuming beverages sweetened with glucose, fructose, sucrose, high fructose corn syrup (HFCS) or aspartame for 2 weeks on the changes of all-day ad libitum food intake, 24-h plasma leptin concentrations and body weight in young males and females.

## 2. Materials and Methods

### 2.1. Participants

Participant in this study are a subgroup from an NIH-funded investigation in which a total of 187 participants assigned to 8 experimental groups were studied. Other metabolic outcomes have been reported from the various experimental groups [19,20,21,22,23]. The current paper reports the results from 131 adults (normal and overweight 18–35 kg/m^2^, males and females, 18–40 yrs) who were assigned to consume 3 sweetened beverages (SB)/day containing either aspartame (*n* = 23) or 25% of their energy requirement (Ereq) as glucose (*n* = 28), fructose (*n* = 28), sucrose (*n* = 24) or HFCS (*n* = 28) for 2 weeks. The experimental protocol was approved by the University of California, Davis (UCD) Institutional Review Board and is registered with Clinical Trials.gov: NCT01103921.

To assess eligibility, potential participants were recruited through an Internet listing (craigslist.com) and local postings of flyers and underwent telephone and in-person interviews with assessments of medical history, complete blood count, and serum biochemistry panel. Inclusion criteria included age 18–40 years and BMI 18–35 kg/m^2^ with a self-reported stable body weight during the prior 6 months. Exclusion criteria included diabetes (fasting glucose >125 mg/dL), evidence of renal or hepatic disease, fasting plasma triglyceride >400 mg/dL, hypertension (>140/90 mm Hg), hemoglobin <8.5 g/dL, and surgery for weight loss. Individuals who smoked, habitually ingested >2 alcoholic beverages or sugar-SB/d, exercised >3.5 h/wk at a level more vigorous than walking, or used thyroid, lipid-lowering, glucose-lowering, antihypertensive, antidepressant, or weight loss medications were also excluded. Assignment to the experimental groups was not randomized; by design, the experimental groups were matched for sex, BMI, and concentrations of fasting insulin, triglyceride, LDL-cholesterol, and HDL-cholesterol in serum collected during the in-person interviews.

For the 5 weeks before study, scheduled participants were asked to limit their daily consumption of sugar-containing beverages to one 8-oz serving of 100% fruit juice and to discontinue consumption of any vitamin, mineral, dietary, or herbal supplements.

### 2.2. Study Protocol

This was a parallel-arm, double-blinded diet intervention study with 3 phases (Figure 1): (1) a 3.5-d inpatient baseline period (Week 0) during which subjects resided at the University of California Davis Clinical and Translational Science Center’s Clinical Research Center (CCRC), consumed a standardized diet, and participated in experimental procedures; (2) a 12-d outpatient intervention period during which subjects consumed their assigned sweetened beverages providing 0% (aspartame-sweetened, no-sugar control) or 25% Ereq as glucose, fructose, sucrose, or HFCS along with their usual ad libitum diets; and (3) a 3.5-d inpatient intervention period (Week 2) during which subjects resided at the CCRC and consumed standardized diets that included the sweetened beverages while all experimental procedures were repeated. Inpatient Week 0 and Week 2 procedures during phases 1 and 3 included: Day (1) fasted body weight and ad libitum food intake trials; Day (2) Standardized eucaloric meals; Day (3) 24-h serial blood collections with standardized eucaloric meals.

Beverages were prepared by a designated staff member at the UC Davis Department of Nutrition Ragle Clinical Research Center. Aspartame-containing beverages were prepared using fruit flavored Market Pantry™ drink mix. HFCS-containing beverages were sweetened with HFCS-55 (ISOSWEET^®^ 5500, 55% fructose, 45% glucose; Tate & Lyle, London, UK). Glucose-containing beverages were sweetened with STALEYDEX^®^ crystalline dextrose (Tate & Lyle). Fructose-containing beverages were sweetened with KRYSTAR^®^ crystalline fructose (Tate & Lyle). Sucrose-containing beverages were sweetened with C&H^®^ cane sugar (Domino Foods Inc., Yonkers, NY, USA). All sugar-SBs were formulated as 15% sugar in water (*w*/*w*) and flavored with unsweetened Kool-Aid^©^ drink mix. Participants were blinded to their beverage assignments, as well as CCRC staff and study personnel who interacted with participants or analyzed samples. Voluntary feedback from the participants indicated that they were unable to distinguish between the sugar- and aspartame-SB. The amount (grams) of beverages provided was standardized among the 5 groups and based on Ereq, which was calculated using the Mifflin equation with a 1.5 adjustment for activity level [24]. During the 12-d outpatient phase, participants were instructed to consume one serving of study beverage with each meal along with their usual diet (three beverage servings/day), and to not consume any other sweetened beverages including 100% fruit juice and fruit drinks.

Riboflavin was added to serve as a biomarker to monitor compliance of beverage consumption [20]. Urine samples were collected during twice weekly appointments and urinary riboflavin concentrations were assessed fluorometrically. Subjects were informed that they were being monitored for beverage consumption, but not informed about the method. There were no group differences in urinary riboflavin concentrations, suggesting comparable compliance among the groups. Additionally, fasting urinary riboflavin concentrations following 9 and 13 d of unmonitored beverage consumption were not different from those following one day of monitored beverage consumption during the inpatient Week 2 period at the CCRC [22], suggesting good compliance during the outpatient period.

#### 2.2.1. Body Weight

Fasted body weight was measured during the Week 0 and Week 2 inpatient periods at the CCRC, with hours fasted and subject attire standardized at both weigh-ins.

#### 2.2.2. Ad Libitum Food Intake Trials

Ad libitum food intake trials were conducted on the first day of the Week 0 and Week 2 inpatient periods at the CCRC. Subjects were instructed to fast for 12-h prior to the 07:00-h inpatient check-in and to not consume any alcohol the day prior to check-in. All subjects were served identical breakfast (09:00-), lunch (13:00-), and dinner (18:00-h) buffet-style meals at both visits. They were told to consume as little or as much of the provided meals as they wished and that they could request additional servings of any of the meal components. During Week 0, meals were provided without SB. At Week 2, SB was provided with the meals and consumption of the SB was mandatory. In total, the 3 provided meals contained a minimum of 300% Ereq. The meals were designed to include commonly consumed palatable selections and healthy selections, but not items that were high in added sugar. Breakfast items included oat-based cereal, scrambled eggs, cheese, salsa, turkey sausage, English muffins and crescent rolls. Lunch included macaroni and cheese, turkey and ham sandwich, whole grain and refined grain-based chips and crackers. Dinner included meat lasagna, pasta with vegetables, crescent and bread rolls. Butter, margarine, and peanut butter were available at all meals, along with an ample vegetable crudité with various dips, and subjects’ preferred milk type. Energy and macronutrient intake from the ad libitum food trials were analyzed based on the nutrition information from the food labels or Food Processor^®^ Nutrition Analysis software (ESHA Research Inc., Salem, OR, USA).

#### 2.2.3. 24-h Serial Blood Collections

24-h serial blood samples were collected on Day 3 of Week 0 and Week 2 inpatient periods via intravenous catheter. On Day 2 and Day 3 of the inpatient Week 0 serial blood collection, all subjects consumed standardized eucaloric meals containing 55% Ereq as complex carbohydrate, 30% fat, 15% protein. The eucaloric meals provided during the inpatient Week 2 period included the assigned study beverages and were as identical as possible to Week 0 meals, except for the isocaloric substitution of the sugar-SB for complex carbohydrate. Three fasting blood samples were collected at 08:00-, 08:30-, and 09:00-h. Twenty-nine postprandial blood samples were collected at 30-to-60-min intervals until 08:00-h the following morning. Meals were served at 09:00-, 13:00-, and 18:00-h. All samples were assayed for leptin and insulin with radioimmunoassay (Millipore Inc., St. Charles, MO, USA). Glucose was measured with an automated glucose analyzer (YIS, Inc., Yellow Springs, OH, USA).

#### 2.2.4. Hydrogen Breath Collections

Consumption of large amounts of pure fructose can lead to fructose malabsorption resulting in adverse gastrointestinal symptoms [25], which can possibly affect energy intake and weight gain [26]. Therefore, to index fructose malabsorption, breath samples were collected before and after beverage consumption and analyzed for hydrogen breath concentration [27]. Participants were asked to consume a low fiber diet during the 8th day of Week 2 and to fast 12-h prior to arrival at the CCRC on the 9th day of intervention (Study Day 12). Prior to the first breath collection, subjects rinsed their mouths with an antibacterial mouthwash. Breath samples, consisting of one normal exhalation into single-patient breath collection bags (QuinTron Instrument Company, Inc), were obtained from each subject before and 30-, 60-, 90-, and 120-min after they consumed their assigned beverage. Hydrogen concentrations were analyzed on the day of collection using the BreathTracker H2 (QuinTron Instrument Company, Inc. Milwaukee, WI, USA).

#### 2.2.5. 24-h Recall of Outpatient Dietary Intake

To determine free living dietary intake, self-administered 24-h dietary intake recalls were collected using the Automated Self-Administered 24-h (ASA 24-h) Dietary Assessment Tool during Week 0 and Week 2 [28]. Subjects entered their food and beverage intake from the previous day into the ASA 24-h food recall site using a study computer at the CCRC on the first day of each inpatient period. For the intervention recall, subjects were instructed to exclude their experimental beverages from the dietary entry and the kcals contained in the experimental beverages were added by staff at a later date.

#### 2.2.6. Physical Activity

Activity during the inpatient periods was monitored and controlled. During Day 1 and 2 of each inpatient period, all subjects participated in a 20-min walk on the CCRC grounds with a staff member. During the days of the 24-h serial blood collections, all subjects were sedentary, and dietary intake was adjusted to account for lower levels of activity. During the outpatient period, all subjects were instructed to continue their normal activity and to not introduce new exercise or workout routines. Subjects were asked to fill out modified versions of the Baecke Physical Activity Questionnaire [29] at the CCRC screening visit and at the end of outpatient Week 2. The screening visit questionnaire asked subjects to describe their general physical activity routine, while the Week 2 questionnaire asked subjects to describe their physical activity during the prior 2 weeks. The questionnaires were analyzed to quantify physical activity (sports, workouts, exercise programs, biking and/or walking to and from work and/or classes) in hours/week. The intensity of these activities was not captured by the questionnaire.

### 2.3. Assays and Statistical Analysis

Fasting leptin was calculated as the mean of the 3 fasting samples. The incremental 24-h leptin area under the curve (AUC) over the morning nadir (lowest value from 09:30- and 11:30-h) was calculated by the trapezoidal method. The AUCs for leptin are therefore expressed as units above each subject’s nadir over 24-h. The change in the 24-h peak over the 24-h nadir was also assessed as this was reported to be the leptin outcome most associated with the change in weight during a 12-week ad libitum high-carbohydrate diet [14]. The incremental 24-h glucose and insulin AUC were calculated over fasting levels by the trapezoidal method. All leptin, glucose, and insulin data were corrected to the in-house standardized QC. The intra- and inter- assay CVs for our laboratory QC were as follows: glucose: 3.6%, 4.5% (intra-assay, inter-assay); insulin: 6.5%, 7.6%; leptin: 8.6% and 16.4%.

The change in hydrogen breath was calculated by subtracting the 0-h value from the peak value during the 2-h collection period.

Percent energy compensation (%EC) was calculated as follows [30]:%EC = [(EI_Week 0_ − EI_Week 2_)/sugar-SB_kcals_] ∗ 100

In this equation, energy intake (EI) represents ad libitum food energy (kcals) intake at Week 0 and Week 2 excluding sugar-SB kcals at Week 2. A value of 100% indicates complete compensation for the provided sugar-SB kcals and values < 100% indicate partial compensation.

Week 0 and Week 2 values were log transformed when the Week 0 values were not normally distributed. Outcomes were analyzed as the absolute (∆) or percent change (%∆) from Week 2 compared with Week 0 using a 2-factor (SB, sex) general linear model (SAS 9.4, SAS, Cary, NC, USA) adjusted for sex*SB, BMI or body fat percentage, and outcome at Week 0 (for absolute ∆ outcomes). Covariates or the interaction term were removed if they did not improve the sensitivity of the model. Significant differences between groups were identified using Tukey’s multiple-comparisons test. Outcomes that were significantly affected within group were identified as least squares mean (LS mean) of the change significantly different than zero. Log transformation failed to normalize the distribution of hydrogen breath concentrations at 0-h or pre-study physical activity; therefore, the change in these outcomes was analyzed by nonparametric tests in PROC NPAR1WAY (SAS 9.4) and by nonparametric *t*-tests in GraphPad Prism 8. Statistical significance was considered at *p* < 0.05. Data are reported as mean ± SD in Table 1, while all other data are mean ± SEM.

## 3. Results

A total of 144 participants were enrolled and assigned to consume aspartame-, or 25% of Ereq as glucose-, fructose-, HFCS-, or sucrose-SB, and 131 completed the study and were included in the analyses: aspartame-SB (*n* = 23), glucose (*n* = 28), fructose (*n* = 28), sucrose (*n* = 24), and HFCS (*n* = 28). There were no significant differences in Week 0 anthropomorphic or metabolic parameters between experimental groups (Table 1). Group means for body weight, ad libitum food energy intake and leptin, glucose, and insulin AUCs at Week 0 and at the end of Week 2 are presented in Table 2, along with the *p*-values for the effects of SB in the ANCOVA models.

### 3.1. Body Weight

The mean absolute change in body weight at Week 2 compared with Week 0 in each group is presented in Figure 2. While the differences among groups were not significant (*p* = 0.092; Table 2), subjects consuming glucose- and HFCS-SB exhibited a significant increase in body weight at Week 2 compared with Week 0 (Glucose: +0.5 ± 0.2 kg, *p* = 0.018; HFCS: +0.8 ± 0.2 kg, *p* = 0.0008). Body weight was not increased in subjects consuming fructose (+0.1 ± 0.2 kg, *p* = 0.79).

### 3.2. Ad Libitum Food Intake Trials

Identical buffet meals were presented at both the Week 0 and Week 2 ad libitum food intake trials. The beverages were included with the Week 2 buffet meals and consumption of the beverages was required. Figure 3A presents the percent compensation for beverage energy, with 100% representing full compensation, thus no increase in energy intake during the Week 2 trials compared to the Week 0 trials. All groups consuming sugar-SB failed to compensate completely for beverage energy. Figure 3B presents the results as % change in energy intake by group at the Week 2 ad libitum food intake trial compared with the Week 0 trial. Subjects consuming fructose-, HFCS-, and sucrose-SB exhibited a significant increase in energy intake compared with subjects consuming aspartame-SB (*p* = 0.037, *p* = 0.039, *p* = 0.0012, respectively). The within-group increase in energy intake was significant in subjects consuming fructose-, HFCS- and sucrose-SB.

Table 3 presents the changes of breakfast, lunch, and dinner energy (kcal) and macronutrient (g) intake during Week 2 compared with Week 0 ad libitum food intake trials not including SB. Subjects consuming sugar-SB partially compensated for the beverage energy consumed during Week 2 by decreasing intake of protein, fat and carbohydrate. There were no group differences in the compensation patterns except for subjects consuming sucrose who had a significant increase in breakfast kcals compared with subjects consuming glucose-SB (*p* = 0.036) and in breakfast carbohydrate compared with subjects consuming glucose- and fructose-SB (*p* = 0.008, *p* = 0.030 respectively). Importantly, fiber consumption, which would have indicated trends toward making more or less healthy food selections (more or less whole grain and vegetable), was unchanged in all groups.

### 3.3. Leptin

Sugar significantly affected 24-h leptin AUC (Figure 4), but not fasting leptin concentrations (*p* = 0.66). Subjects consuming fructose had a significant decrease in 24-h leptin AUC compared with Week 0 (−13.6 ± 7.6 ng/mL × 24-h, *p* = 0.0006) and compared with subjects consuming sucrose (*p* = 0.0008). Only subjects consuming sucrose-SB exhibited an increase in 24-h leptin AUC compared with Week 0 (25.0 ± 8.9 ng/mL × 24-h, *p* = 0.025). This increase was driven mainly by the females consuming sucrose (females: 44.9 ± 12.0 ng/mL × 24-h, *p* = 0.0056 vs. Week 0; males: 6.2 ± 11.4 ng/mL × 24-h, *p* = 0.89 vs. Week 0). The change in amplitude (the 24-h peak over the 24-h nadir) at Week 2 compared with Week 0 was also significantly affected by sugar (Table 2), with decreased amplitude induced by fructose (−0.94 ± 0.47 ng/mL, *p* = 0.0019 vs. Week 0) and increased amplitude induced by sucrose (1.5 ± 1.2 ng/mL, *p* = 0.044 vs. Week 0). The changes in leptin amplitudes were also affected by sex with significant increases in females (0.30 ± 0.61 ng/mL) compared with males (−0.002 ± 1.5, *p* = 0.0003 ng/mL females vs. males). Neither the 24-h leptin AUC nor the change in amplitude were significantly associated with changes of body weight (leptin AUC: r^2^ = 0.0003, *p* = 0.84; leptin amplitude: r^2^ = 0.0001, *p* = 0.89, simple regression) or energy intake (leptin AUC: r^2^ = 0.010, *p* = 0.25; leptin amplitude: r^2^ = 0.006, *p* = 0.39, simple regression) during the ad libitum food intake trials.

### 3.4. Glucose and Insulin

The glucose and insulin 24-h AUCs at Week 0 and Week 2 with effect of SB *p*-value are shown in Table 2. Subjects consuming fructose had a significant decrease in glucose 24-h AUC compared with Week 0 (−82.9 ± 16.4 mmol/L, *p* < 0.0001) and compared with subjects consuming aspartame, glucose, HFCS, and sucrose (*p* = 0.0023, *p* < 0.0001, *p* = 0.021, *p* = 0.0002). Subjects consuming glucose had a significant increase (+111.7 ± 16.5 mmol/L, *p* < 0.0001) compared with Week 0 and compared with subjects consuming aspartame, fructose, HFCS, and sucrose (*p* = 0.0005, *p* < 0.0001, *p* < 0.0001, *p* = 0.0037).

Subjects consuming fructose had a significant decrease in 24-h insulin AUC compared with Week 0 (−147.2 ± 33.9 pmol/L, *p* < 0.0001) and compared with subjects consuming aspartame, glucose, HFCS, and sucrose (*p* = 0.018, *p* < 0.0001, *p* = 0.044, *p* < 0.0001). There was a significant increase in 24-h insulin AUC in subjects consuming glucose compared with Week 0 (+167.6 ± 33.9 pmol/L, *p* < 0.0001) and compared with subjects consuming aspartame, fructose, and HFCS (all *p* < 0.001).

### 3.5. Hydrogen Malabsorption

Figure 5 presents the hydrogen breath response indexed as the peak value over 2-h minus the 0-h value immediately before SB consumption. Fasting hydrogen breath values, which represent the 0-h values, did not differ among the five groups (*p* = 0.56, Kruskal–Wallis Test). Following the consumption of one serving of assigned beverage, subjects consuming fructose-SB had significantly higher increases in peak hydrogen breath concentrations compared with the 0-h value and compared with all other SB group.

### 3.6. Outpatient 24-h Dietary Intake and Physical Activity

A subset of 62 participants completed both the Week 0 and Week 2 dietary recalls describing free living intake. Subjects reported consuming significantly less energy at both Week 0 (−29 ± 38(SD)%, *p* = 1.9 × 10^−9^) and Week 2 (−12 ± 36%, *p* = 0.004) than their calculated Ereq. The deficits in reported energy intake compared with calculated Ereq, which were likely due to under-reporting [31] and/or undereating, were higher at Week 0 than Week 2 (*p* = 0.02). The mean change (Week 2 minus Week 0) in reported energy intake ± SD for all 62 subjects was +366 ± 1324 kcals with a range of −3539 to +3979 kcals. This change was significantly affected by SB group (F statistic = 3.19, *p* = 0.02), with a significant increase in energy intake reported by subjects consuming sucrose compared with that reported by subjects consuming aspartame (*p* = 0.01). However, the effect of SB on the change in reported energy intake was minimal compared with the effect of underreporting/undereating at Week 0, included in the statistical model as reported energy intake at Week 0/calculated Ereq (*p* < 0.0001; F statistic = 54.9). Specifically, subjects who showed the greatest level of underreporting/undereating at Week 0 had the greatest increases in reported energy intake at Week 2. This confounding by underreporting/undereating at Week 0, as well as the limited sample size and number of recalls, preclude drawing conclusions regarding the effects of SB group on outpatient energy intake.

The amount of physical activity subjects reported as their usual routine during the pre-study period (4.5 ± 3.8 (SD) hours/week) was not different among SB groups (*p* = 0.10, Kruskal–Wallis test). Subjects reported engaging in significantly less physical activity during the 2-week intervention period than during pre-study (−0.80 ± 3.4 (SD) hours/week, *p* = 0.0047, Wilcoxon matched-pairs signed rank test). This decrease was not affected by SB group (*p* = 0.27, Kruskal–Wallis Test). None of the main outcomes (change in body weight and 24-h leptin, glucose, and insulin AUC) were affected by the reported pre-study level of physical activity nor by the change in physical activity.

## 4. Discussion

This study investigates the effects of consuming glucose-, fructose-, HFCS-, sucrose-, and aspartame-SB for 2 weeks in 18–40-year-old males and females, with BMIs ranging from 18 to 35 kg/m^2^, on body weight, ad libitum food intake from breakfast, lunch and dinner buffets, and 24-h leptin response.

Body weight gain due to fructose consumption is conflicting and inconclusive. Recent meta-analyses using controlled feeding trials found no effect of fructose consumption on body weight when compared with isocaloric amounts of carbohydrate and when ≤100 g is consumed [32,33]. However, another systematic review and meta-analysis found that, among individuals consuming ad libitum diets, the intake of free sugar, which included fructose and fructose-containing sugars, was a determinant of body weight [34]. We hypothesized that the failure of fructose to stimulate leptin secretion could be a mechanism by which fructose and fructose-containing sugar could promote increased food intake and body weight gain. Our previous study did not support this hypothesis. Body weight gain was comparable in older, overweight and obese males and females after 8 weeks of consuming fructose- or glucose-SB with their usual ad libitum diets [18], even though 24-h leptin concentrations were significantly decreased by fructose consumption [17]. However, it is possible that individuals who are younger and/or leaner are more responsive to changes in circulating leptin.

As with our previous study, subjects consuming fructose-SB exhibited the expected decrease in 24-h leptin AUC, but this decrease was not linked to body weight gain. In fact, counter to our hypothesis, the subjects consuming fructose exhibited the least weight gain over the 2 weeks of study (~0.1 kg) among the four groups consuming sugar-SB. This result may possibly be explained by the suggestion that leptin evolved to minimize weight loss during periods of negative energy balance; thus, decreases or increases in circulating leptin have little effect on body weight during neutral or positive energy balance [35]. However an additional or alternative explanation, which has not been widely considered, for the failure of the lowered leptin levels to promote body weight gain in the subjects consuming fructose, may be decreased energy availability due to fructose malabsorption [36]. Consumption of fructose as a monosaccharide can overwhelm the absorptive capacity of the small intestine leading to fructose malabsorption, and the resulting presence of the fructose in the large intestine can be indexed as breath hydrogen concentrations [25]. Our data show that subjects consuming fructose-SB had significantly higher peak hydrogen breath concentrations over the 2-h following beverage consumption compared with all of the other SB groups. It is also possible that fructose malabsorption limits energy intake due to causing gastrointestinal manifestations including diarrhea, bloating, and abdominal distention. Indeed, the number of reports of these types of adverse effects were higher in the group consuming fructose-SB compared to any of the other groups.

Subjects consuming fructose did consume more calories during the Week 2 ad libitum food intake trial when beverages were consumed. However, this was unlikely to be a response mediated by their lowered leptin concentrations since all groups consuming sugar-SB significantly increased, or tended to increase, energy intake when consuming their assigned sugar-SB with the ad libitum buffet meals. Furthermore, counter to our hypothesis, the sucrose-SB group had the highest increases in energy intake during these trials, yet they also exhibited the highest increases in 24-h leptin AUC and leptin amplitudes.

We did not expect to observe the highest increases of 24-h leptin concentrations in the subjects consuming sucrose-SB. Leptin secretion by adipocytes is mediated by insulin-mediated glucose metabolism [11,37]; thus, we expected the leptin responses to parallel the glucose and insulin responses. Since the subjects consuming glucose had higher increases in glucose and insulin 24-h AUC than all other groups, we expected this group to exhibit the highest increases in leptin. We cannot explain why this did not occur, nor why 24-h leptin AUC was significantly increased in subjects consuming sucrose-SB, specifically the females.

The data suggest that fructose is the principal driver of the adverse metabolic outcomes induced by added sugar consumption [38]. However, the current data show that regardless of sugar type and fructose content, sugar-SB can increase the risk of weight gain due to a lack of complete compensation for the energy contained in the sugar-SB. These results corroborate the results of Kuzma et al., who conducted two randomized, controlled, eight-day crossover studies in which adults consumed: Study (1) fructose-, glucose-, and aspartame-SB; Study (2) fructose-, glucose-, and HFCS-SB [39]. Standardized meals containing 125% Ereq were provided daily during all arms of both studies, and the sugar-SB contained an additional 25% Ereq. Energy intake increased during sugar-SB consumption, regardless of sugar type, and it did not increase during the consumption of aspartame-SB. Importantly, for our study and the studies by Kuzma et al., the increases in energy intake did not correlate with changes of body weight. Clearly, longer-term, randomized, controlled trials assessing the effects of sugar-SB consumption on ad libitum energy intake and body weight are still needed.

The current results and the results of Kuzma et al. also pertain to the ongoing controversy about the effects of non-nutritive sweeteners on body weight. There is epidemiological evidence that associates the consumption of non-nutritive sweeteners with positive energy intake and body weight gain [40,41,42]. With about 41% of adults and 25% of children in the US actively consuming these sweeteners, this evidence is highly relevant to the obesity epidemic [43]. However, the results from the current study and numerous other clinical dietary intervention studies [44] demonstrate without exception that providing aspartame-SB with ad libitum meals does not promote increased energy intake and/or body weight gain [45]. Nevertheless, aspartame consumption has decreased by 10% since 2002 [46], and a GlobalData© market research report noted that consumers perceive aspartame as having an even more negative impact on health than HFCS [47]. These findings could be influenced by online lay media, both scholarly [48] and sensational [49], that contradict the evidence from the dietary intervention studies [44] and conclude that aspartame promotes increased appetite, energy intake and obesity.

Our results demonstrate that changes in circulating leptin did not influence body weight gain or ad libitum food intake in young males and females consuming sugar-SB for 2 weeks. Thus, the failure of these subjects to completely compensate for beverage energy during our ad libitum food intake trials may have been driven by failure to elicit appropriate hormonal response of glucagon-like peptide, gastric inhibitory polypeptide, peptide YY or ghrelin, and/or by effects of sugar-SB on hedonic drive. More studies are needed to determine the specific mechanisms mediating increased energy intake in subjects consuming sugar-SB.

### Strengths and Limitations

To our knowledge, this is the first study to investigate all-day ad libitum food intake and 24-h leptin concentration in young adults before and after they consume beverages containing aspartame or 25% of Ereq as glucose, fructose, HFCS, or sucrose for 2 weeks.

By design this study was not randomized. The subject assignment to experimental groups; based on sex, BMI, and concentrations of fasting triglyceride, LDL-cholesterol, HDL-cholesterol and insulin; could have introduced a potential bias. Subjects were provided with only three dietary guidelines during the 12-day outpatient period: (1) experimental beverage must be consumed with each of the three main meals; (2) except for the experimental beverages, no sweet/sweetened beverages, including 100% fruit juice and fruit drinks, may be consumed; (3) no alcohol may be consumed the days before CCRC check-in. Therefore, variations in the outpatient diets that the subjects consumed on the days prior to the ad libitum food intake trial may have influenced the results. Moreover, since sugar-containing solid foods were not restricted, we do not know the total amount of sugar that was consumed by each subject during the outpatient period. We attempted to address these limitations by collecting 24-h dietary intake recalls from some of the participants. However, the 24-h dietary recall data were confounded by a limited sample size, number of completed recalls, and by under-reporting and/or undereating; thus, they were not valid for providing insights on outpatient food intake or on the effects that the sugars may have on energy intake in free-living subjects [50]. The 16-day intervention period was relatively short, and the changes of body weight may not correlate to what would be observed over a longer period.

## 5. Conclusions

The addition of sugar-SB, but not aspartame-SB, to ad libitum meals led to increased energy intake. These results suggest that consuming sugar-SB, regardless of the type of sugar, leads to failure to fully compensate for energy consumed as sugar-SB and could ultimately contribute to weight gain. The results do not support our hypothesis that effects of sugar-SB consumption on energy intake and body weight gain are mediated by the fructose-induced reduction in circulating leptin. Energy intake and weight gain in subjects consuming fructose-SB may be influenced by fructose malabsorption.

## Figures and Tables

**Figure 1 nutrients-12-03893-f001:**
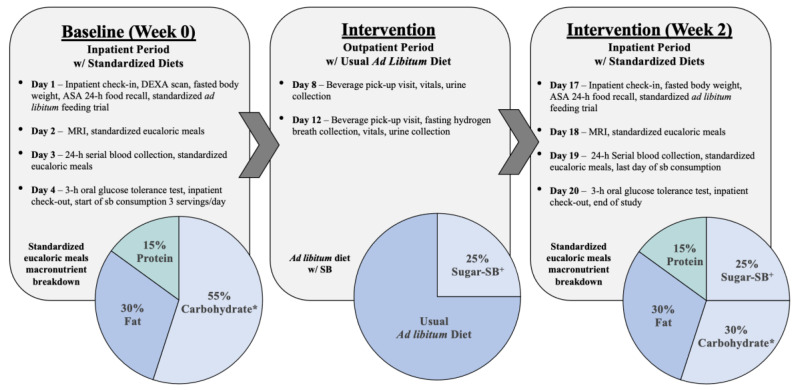
Study design, experimental testing days, and dietary protocol. Sweetened beverage (SB), dual energy X-ray absorptiometry (DEXA), automated self-administered 24-h (ASA 24-h) dietary assessment tool, high fructose corn syrup (HFCS). * <2% added sugar. ^+^ Either 0% energy requirement (Ereq) as aspartame or 25% Ereq as glucose, fructose, HFCS or sucrose.

**Figure 2 nutrients-12-03893-f002:**
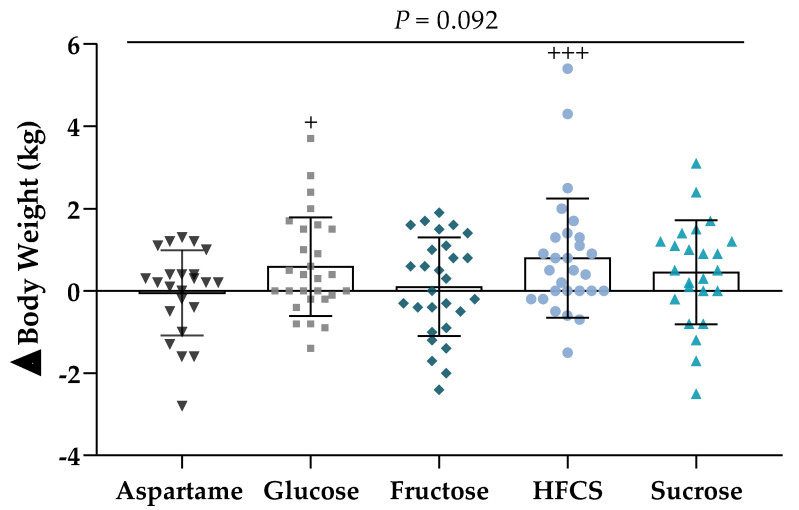
Body weight: The mean ± SEM of the absolute change (Week 2–Week 0) in body weight in subjects consuming either glucose-, fructose-, high fructose corn syrup (HFCS)-, sucrose-, or aspartame-sweetened beverages for 2 weeks. Two-factor (SB group, sex) analysis of covariance with adjustment for outcome at Week 0; ^+^
*p* < 0.05, ^+++^
*p* < 0.001, least squares (LS) mean different from zero.

**Figure 3 nutrients-12-03893-f003:**
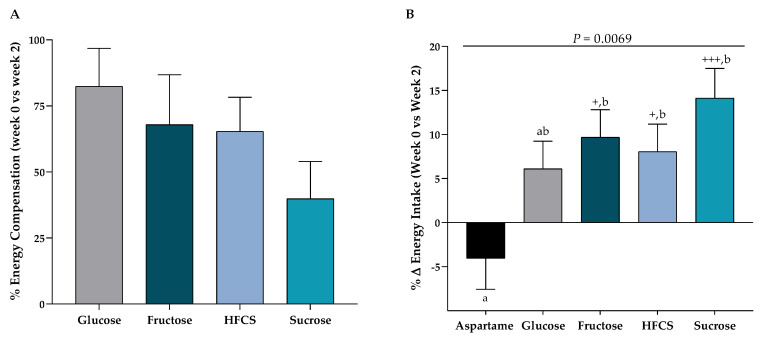
Energy compensation and intake during ad libitum food intake trials: Subjects consumed 3 ad libitum buffet meals at Week 0 and consumed identical meals with glucose-, fructose-, high fructose corn syrup (HFCS)-, sucrose-, or aspartame-sweetened beverages at Week 2. (**A**) Mean ± SEM of the percent energy compensation for sugar-SB at Week 2 compared with Week 0 with < 100% denoting partial compensation. (**B**) Mean ± SEM of the % change in energy intake at Week 2 compared with Week 0. *p* = 0.0069 effect of SB, 2-factor (SB group, sex) analysis of covariance with adjustment for BMI; ^+^
*p* < 0.05, ^+++^
*p* < 0.001, LS mean different from zero; a different from b, Tukey’s.

**Figure 4 nutrients-12-03893-f004:**
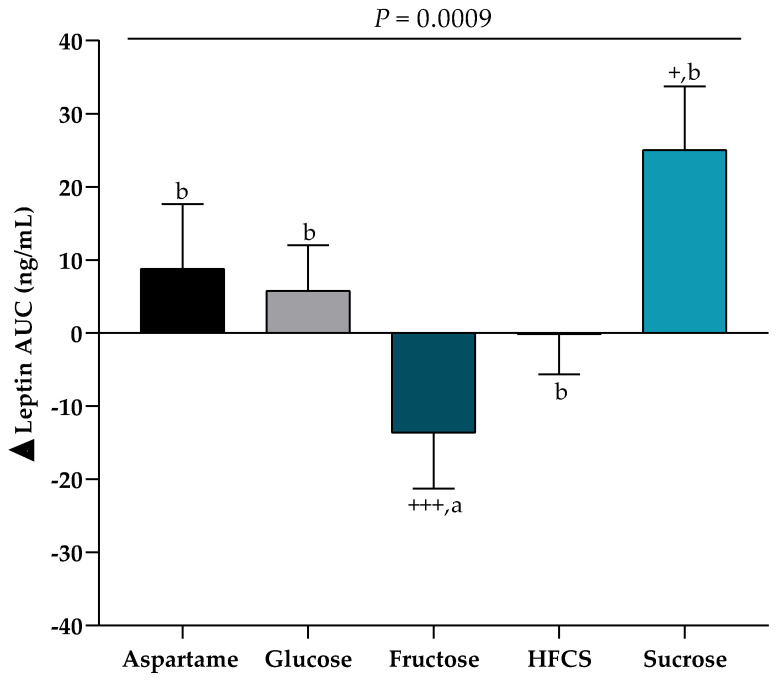
24-h Leptin AUC: The mean ± SEM of the absolute change (Week 2–Week 0) in leptin AUC in subjects consuming either glucose-, fructose-, high fructose corn syrup (HFCS)-, sucrose-, or aspartame-SB for 2 weeks. *p* = 0.0009 effect of SB, 2-factor (SB group, sex) analysis of covariance with adjustment for % body fat and outcome at Week 0; ^+^
*p* < 0.05, ^+++^
*p* < 0.001, LS mean different from zero; a different from b, Tukey’s multiple comparison test.

**Figure 5 nutrients-12-03893-f005:**
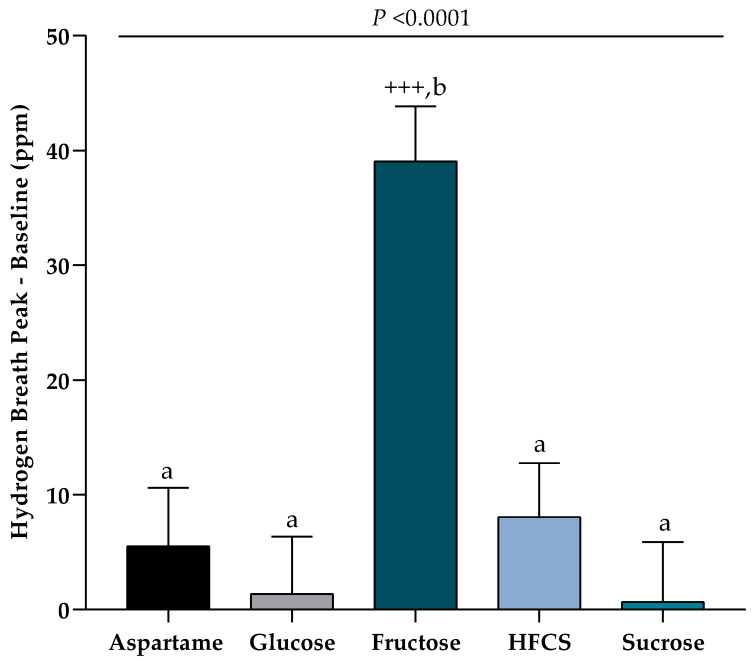
Hydrogen breath after consuming SB: The change in peak hydrogen breath–0-h value in subjects consuming glucose-, fructose-, high fructose corn syrup (HFCS)-, sucrose-, or aspartame-SB for 2 weeks. *p* < 0.0001 effect of SB, Kruskal–Wallis test; ^+++^
*p* < 0.001, Wilcoxon matched-pairs signed rank test; a different from b, Mann–Whitney unpaired tests for each SB versus fructose.

**Table 1 nutrients-12-03893-t001:** Description of participant characteristics at Week 0 by sugar-sweetened beverage group ^1^.

Parameter	Aspartame	Glucose	Fructose	HFCS	Sucrose
Age (year)	25.4 ± 6.2	26.0 ± 5.74	26.8 ± 6.2	26.8 ± 6.6	25.9 ± 6.3
Sex (M/F)	11/12	15/13	15/13	15/13	12/12
Weight (kg)	71.8 ± 10.6	75.5 ± 12.8	75.7 ± 12.9	72.8 ± 14.5	71.9 ± 12.1
BMI (kg/m^2^)	24.8 ± 3.3	25.8 ± 3.5	25.4 ± 3.7	24.9 ± 3.9	25.3 ± 3.4
Waist circumference (cm)	75.2 ± 6.4	79.0 ± 9.3	77.8 ± 10.5	76.9 ± 10.1	75.4 ± 7.2
Body fat (%)	27.1 ± 9.6	28.9 ± 8.4	28.9 ± 10.3	26.0 ± 9.7	29.1 ± 11.6
TG (mg/dL)	100.5 ± 52.6	101.6 ± 47.2	99.3 ± 34.9	107.8 ± 50.1	113.6 ± 48.4
Total cholesterol (mg/dL)	148.9 ± 25.5	161.8 ± 31.1	150.6 ± 24.8	157.6 ± 34.3	159.1 ± 23.1
HDL (mg/dL)	39.4 ± 7.4	45.6 ± 14.9	44.4 ± 9.4	45.6 ± 13.7	42.9 ± 6.6
Blood pressure (mm Hg)	69.2 ± 8.6/112.3 ± 11.5	73.8 ± 8.3/118.9 ± 11.2	71.5 ± 6.5/117 ± 9.8	72.2 ± 7.2/117.1 ± 9.9	72.2 ± 5.5/114.3 ± 8.4

^1^ Values are mean ± SD. high fructose corn syrup (HFCS), body mass index (BMI), triglyceride (TG), high-density lipoprotein (HDL).

**Table 2 nutrients-12-03893-t002:** Body weight, energy intake during ad libitum food intake trial and 24-h leptin, glucose, and insulin AUC at Week 0 and Week 2 ^1^.

	Aspartame		Glucose		Fructose		HFCS		Sucrose		*p* Value Effect of SB
	Week 0	Week 2	Week 0	Week 2	Week 0	Week 2	Week 0	Week 2	Week 0	Week 2	
Body Weight (kg) ^1^	71.8 ± 2.2	71.7 ± 2.2	75.5 ± 2.4	76.1 ± 2.5	75.7 ± 2.4	75.8 ± 2.4	72.9 ± 2.7	73.7 ± 2.8	71.9 ± 2.5	72.4 ± 2.6	0.092 *
Ad Lib Meal Energy Intake w/SB (kcal) ^2^	2900 ± 151	2727 ± 129	2910 ± 127	3034 ± 105	2816 ± 155	3005 ± 163	3089 ± 141	3304 ± 155	2793 ± 160	3155 ±178	0.0069 ^†^
Ad Lib Meal Energy Intake w/o SB (kcal)	2900 ± 151	2727 ± 129	2910 ± 127	2426 ± 97	2816 ± 155	2393 ± 161	3089 ± 141	2706 ± 144	2793 ± 160	2567 ± 170	-
Sugar-SB (kcal)	-	-	-	608 ± 15	-	612 ± 15	-	598 ± 17	-	588 ± 17	-
Leptin AUC (ng/mL × 24-h^) 3^	76.0 ± 13.7	84.7 ± 16.4	76.2 ± 9.5	82.0 ± 11.1	77.4 ± 11.2	63.7 ± 9.9	84.3 ± 14.7	83.7 ± 14.9	92.2 ± 17.8	117.3 ± 23.8	0.0009 *
Leptin Amplitude (ng/mL) ^4^	7.1 ± 1.1	7.2 ± 1.3	8.1 ± 2.0	8.3 ± 2.1	8.1 ± 2.0	7.2 ± 2.1	8.1 ± 2.0	8.1 ± 2.1	9.7 ± 2.0	11.2 ± 2.4	0.0035 *
Insulin AUC (pmol/L × 24-h) ^5^	513.4 ± 91.2	484.8 ± 78.4	474.9 ± 52.2	650.2 ± 83.7	568.6 ± 128.8	400.6 ± 65.3	449.1 ± 47.6	429.6 ± 42.7	447.9 ± 55.8	512.8 ± 48.2	<0.0001 *
Glucose AUC (mmol/L × 24-h) ^6^	171.1 ± 27.6	182.7 ± 25.1	155.6 ± 21.1	269.2 ± 20.1	183.6 ± 34.1	93.7 ± 20.2	195.0 ± 23.5	173.8 ± 25.5	144.0 ± 25.5	178.4 ± 27.7	<0.0001 *

^1^ Values are means ± SEM. Area under the curve (AUC), high fructose corn syrup (HFCS), sweetened beverage (SB). * Effects of SB on absolute change of outcome, ^†^ Effect of SB on percent change of outcome. Effect of sugar in the primary 2-factor (sugar, sex) ANCOVA model that included adjustment for BMI ^2,6^, %body fat ^3,4^, outcome at Week 0 ^1–6^.

**Table 3 nutrients-12-03893-t003:** Change in energy and macronutrient intake (Week 2–Week 0) during ad libitum food intake trials ^1^.

	Aspartame	Glucose	Fructose	HFCS	Sucrose
Breakfast (kcals)	−2 ± 62 ^ab^	−13 ± 48 ^b^	58 ± 57 ^ab^	87 ± 42 ^ab^	236 ± 108 ^a^
Fat (g)	−0.2 ± 4.0	−11.9 ± 2.8	−8.0 ± 3.8	−5.8 ± 2.9	−1.5 ± 4.6
CHO (g)	0.8 ± 5.7 ^ab^	−18.7 ± 5.0 ^a^	−13.3 ± 5.5 ^a^	−8.8 ± 3.4 ^ab^	21.3 ± 16.3 ^b^
Fiber (g)	−0.2 ± 0.3	−1.8 ± 0.6	1.9 ± 0.6	0.4 ± 0.4	1.1 ± 1.2
Protein (g)	−0.1 ± 2.6	−9.1 ± 2.3	−6.4 ± 2.9	−7.4 ± 2.4	−4.2 ± 4.2
Sugar-SB (g)	-	55.0 ± 1.3	55.5 ± 1.4	54.1 ± 1.5	53.2 ± 1.5
Lunch (kcals)	−47 ± 58	74 ± 61	51 ± 53	45 ± 40	30 ± 61
Fat (g)	0.1 ± 3.3	−5.8 ± 3.2	−5.8 ± 3.2	−5.8 ± 2.1	−6.3 ± 3.1
CHO (g)	−10.7 ± 6.1	−10.4 ± 5.7	−15.7 ± 4.5	−19.7 ± 4.2	−18.0 ± 6.6
Fiber (g)	−0.3 ± 0.5	−0.7 ± 0.4	−0.3 ± 0.4	−1.2 ± 0.4	−0.2 ± 0.6
Protein (g)	−5.4 ± 2.9	−9.6 ± 2.7	−8.8 ± 2.6	−7.7 ± 2.2	−9.4 ± 2.8
Sugar-SB (g)	-	55.0 ± 1.3	55.5 ± 1.4	54.1 ± 1.5	53.2 ± 1.5
Dinner (kcals)	−64 ± 56	63 ± 58	81 ± 50	77 ± 53	145 ± 37
Fat (g)	−1.0 ± 3.2	−4.8 ± 2.8	−6.3 ± 2.7	−4.5 ± 2.8	−1.4 ± 1.8
CHO (g)	−10.8 ± 4.9	−18.3 ± 7.2	−13.4 ± 5.2	−13.6 ± 6.1	−7.0 ± 4.0
Fiber (g)	−0.8 ± 0.4	−2.8 ± 1.7	−0.9 ± 0.5	−1.4 ± 0.5	0.4 ± 0.6
Protein (g)	−2.3 ± 3.3	−4.2 ± 2.1	−3.4 ± 2.2	−5.3 ± 2.0	−2.0 ± 1.5
Sugar-SB (g)	-	55.0 ± 1.3	55.5 ± 1.4	54.1 ± 1.5	53.2 ± 1.5

^1^ Mean ± SEM. High fructose corn syrup (HFCS), total carbohydrate content from meals (CHO), i.e., excludes sugar-sweetened beverage (SB). ^b^ significantly different from, ^a^ Tukey’s multiple comparison test.

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
