# Peer review of "Effects of Consuming Sugar-Sweetened Beverages for 2 Weeks on 24-h Circulating Leptin Profiles, Ad Libitum Food Intake and Body Weight in Young Adults"

_nutrients, 2020, doi:10.3390/nu12123893_

Round 1

Reviewer 1 Report

The study is clearly written up. The methods are detailed and the results clearly depicted in tables and figures. Excellent referencing,

The abstract should indicate the sample size- otherwise clear.

Author Response

The study is clearly written up. The methods are detailed and the results clearly depicted in tables and figures. Excellent referencing,

  1. The abstract should indicate the sample size- otherwise clear.

We are gratified by the Reviewer’s positive comments. We have adjusted the abstract to include the total sample size on line 28 in addition to the originally stated number of participants per group on line 30:

“In a parallel, double-blinded, intervention study, participants (n=131, BMI 18-35 kg/m2; 18-40yrs) consumed three beverages/day containing aspartame or 25% energy requirement as glucose, fructose, HFCS or sucrose (n=23-28/group).

In addition to our response to your comment, we regret that there were a few errors in the manuscript that needed to be corrected. In Table 2 and the abstract we mistakenly used the p-values for the untransformed data instead of the transformed data for leptin AUC (new P=0.0009 vs. old P=0.0022) & leptin amplitudes (new P=0.0035 vs. old P=0.013). A few untransformed p-values were also incorrectly used in the results sections for the leptin, glucose, and insulin AUC results and have been updated with the transformed p-values. Also in Table 2 and in Figure 3B the p-value for the absolute change in kcal intake (P=0.0032) was used instead of the percent change (P=0.0061). Lastly, Table 3 had two subscripts that were in the wrong row. These corrections did not alter the significance of any outcome.

Reviewer 2 Report

This is a much needed experiment looking at the impact of different sweetener consumption on leptin, insulin, body weight and ad libitum food intake. The methods described are generally sound and the experimental protocol appears rigorous. My main concern is a lack of a few critical variables to include in analyses which may illuminate the results. 

Major comments

The hypothesis states that fructose would fail to stimulate leptin, is this hypothesis a direct comparison of fructose on leptin or via fructose’s known independence from insulin?

Unsure what “Voluntary feedback from the participants indicated that they
were unable to distinguish between the sugar- and aspartame-SB.“ means. Were they given visual analogue scales to indicate they could not tell the difference in sweetness? Were visual analogue scales used to measure hunger or satiety? 

A figure outlining the study timeline and procedures would be helpful

In table 2, it appears that leptin is higher in the HFCS group and sucrose group at baseline. Is this accurate? Were the baseline differences meaningful?

Figure 1 could be more informative, would be nice to see the distribution of the data points either with a violin plot or showing the data points on the existing plot

Was any measure of body fat used? Since leptin hormone produced in adipose tissue this would be an important consideration

Were lipoproteins or plasma lipids measured at multiple time points? I see baseline values but not at the 2-week mark

Were only two dietary recalls performed? Standard diet recall procedure is 3 recalls per time point of interest (6 recalls in total). It is hard to interpret the current diet recalls as they don’t appear to reflect actual consumption. Is it possible to get the participants to fill out the ASA 3 more times to get a more accurate picture of habitual dietary intake?

You note that, “We cannot explain why this did not occur, nor 24-h leptin AUC was significantly increased in subjects consuming sucrose-SB, specifically the females “ This could be potentially explained via body fat mass if that was measured. Or via lipoprotein or lipid profiles. Alternatively, adiponectin could also be an important factor if that was measured.

The authors mention leptin as a “satiety signal” however, increasingly it is noted as a marker of body weight status and not short term satiety. Were any of the gut peptide hormones which have show a more direct impact on satiety measured such as ghrelin (in terms of decreasing satiety)?

Minor comments

Font on figure 2 is difficult to read

Table 3 is difficult to interpret the meaning of the a b and c. Is there a more direct way to indicate the contrasts used?

Passive sentence needs re-phrase “There are much data to suggest that fructose is the principal driver of the adverse metabolic  outcomes induced by added sugar consumption“

Reviewer 3 Report

Introduction

  1. Please describe more details about the blunted satiety signaling and brain reward pathway.
  2. Why didn't research focus on the effects of fructose induction? What was the role of aspartame in this study?

Methods

  1. Why the study set the sweetened beverage provides 25% of total calories?
  2. P6, line 232. What is the acronymous of LS?

Results

  1. P6, line 241-242. Please check the hydrogen breath data of the Glucose group. The value was lower than in other groups.

Discussion

  1. Please explain why fructose consumption makes the body weight gain.
  2. Does fructose consumption affect leptin signaling? What kind of mechanism can support this issue?

Round 2

Reviewer 2 Report

Appreciate the efforts to address comments. Acceptable to publish once edits are incorporated.